# The Geotectonic Peculiarities of the North Caspian Permian Salt-Bearing Basins (Kazakhstan)



Vyacheslav Zhemchuzhnikov, Aitbek Akhmetzhanov *[ID], Kenzhebek Ibrashev and Gauhar Akhmetzhanova

School of Geology, Kazakh-British Technical University, Almaty 050000, Kazakhstan;
v.zhemchuzhnikov@kbtu.kz (V.Z.); rectorkbtu@kbtu.kz (K.I.); g.akhmetzhanova@kbtu.kz (G.A.)
* Correspondence: a.akhmetzhanov@kbtu.kz; Tel.: +7-701-764-8906

**Abstract:** This article examines the geotectonic and sedimentary features of the Upper Devonian–Carboniferous–Permian deposits of the North Caspian basin, represented by deposits of marine Paleozoic-isolated carbonate platforms formed during the subsidence of the basement on the passive continental margin. The top is covered by thick salt-bearing Kungurian deposits from the end of the Early Permian epoch. The formation of carbonate platforms is associated with a major tectonic restructuring of the basin at the turn of the Caledonian and Hercynian eras, when the Paleo-Tethys Ocean was formed and isolated carbonate islands began to grow in an open marine environment. The central part of the depression experienced a long and gradual subsidence that spanned the entire Paleozoic era and the beginning of the Mesozoic era. In the south and east, from the Devonian to the Permian periods, barriers were formed in the form of island carbonate massifs that separated the North Caspian basin from the Paleo-Tethys Ocean. During the formation of the salt-bearing basin, these barriers limited water exchange and ensured a one-way influx of sea water from the open ocean. As a result, at the end of the Permian period, thicker salts accumulated; however, during the collision of the continental massifs, an invasion of many kilometers of redbeds occurred. They initially stopped salt accumulation; however, gradually, in the north of the Caspian Sea during Roadian times, the salt accumulation continued. The post-Roadian time is associated with the influx of large quantities of redbed sediments, which caused gravitational instability in the underlying salt, and salt tectonics began with the formation of domal structures.

**Keywords:** North Caspian basin; carbonate platform; Kungurian salt; Paleozoic; Tethys; evolution

## 1. Introduction

The North Caspian basin, or with its local name, "Pre-Caspian depression", is a unique salt-bearing basin located in the west of Kazakhstan (Figure 1). Its modern appearance is located in the contour of the salt dome region. The western and northwestern boundary of the depression is the pre-Kungurian tectono-sedimentary side of the south-eastern end of the East European Platform, stretching from north to south along the longitude through the city of Saratov to the city of Volgograd and further away. Also, it turns out to the east and then runs along the latitude of Uralsk city and on to the city of Orenburg. From the east, the depression is limited by the Uralian folded structures and from the southeast by the western closure of the South Emba high. In the south and southwest, this planetary structure is closed by the folded system of the Donbass–Tuarkyr uplifts [1,2]. The formation of the depression began at the end of the Early Permian epoch, at the final stage of the Uralian orogeny, when, as a result of the collision of the East European and Siberian paleocontinents, as well as the Kazakhstania paleocontinent and smaller Central Asia continental blocks, the closure of the Uralian–Mongolian paleocean (Paleo-Tethys) occurred. On the site of the modern Caspian Sea, a marine basin was formed, detached from the gulf-shaped Paleo-Tethys Ocean forming along the eastern coast of the supercontinent Pangea [3–5]. The shape and size of those basins almost coincided with the modern North Caspian basin, and

it connected through a narrow strait with the Paleo-Tethys Ocean. Located in low tropical paleolatitudes, it became the place where the intensive accumulation of evaporites began.

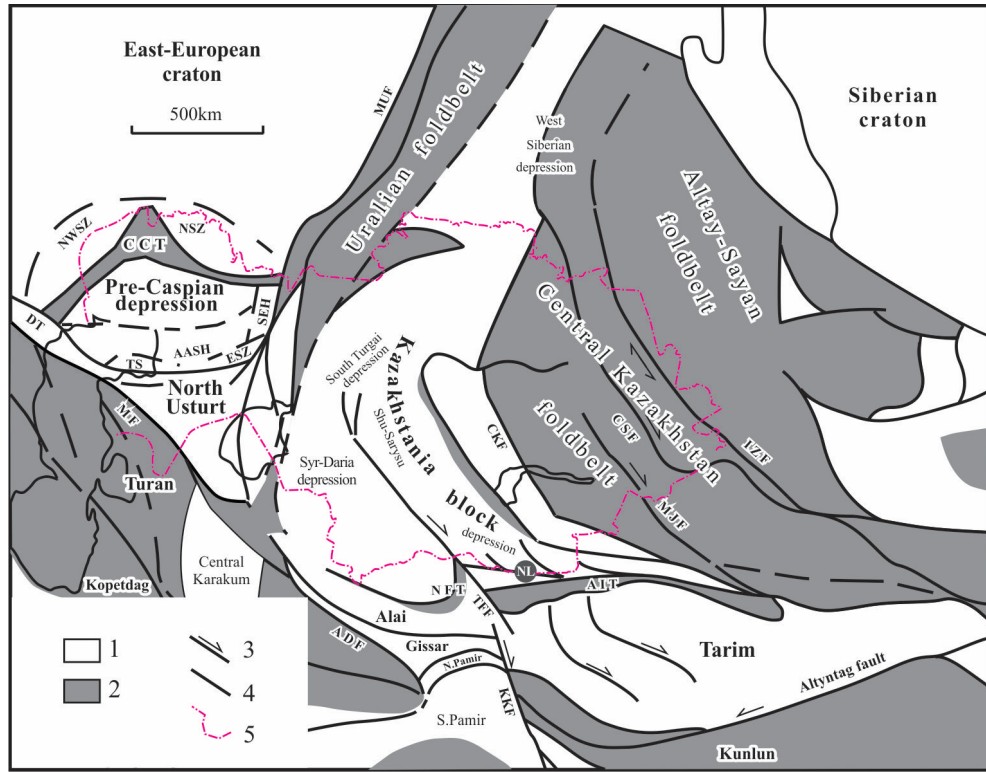

**Figure 1.** Regional schema of position of the North Caspian basin in the system of the main tectonic structures of Kazakhstan and adjacent regions of Central Asia, the East European and Siberian platforms (according to [6], with changes and additions). Abbreviations: IZF—Intysh–Zaisan fault; SCF—Central Shyngyz fault; MJF—main Dzhungar fault; CKF—central Kazakhstan fault; JNF—Jalair-Naiman fault; MKF—main Karatau fault; TFF—Talasso—Fergana fault; KKF—Karakorum fault; ADF—Amu Darya fault; MF—Mangistau fault; MUF–main Ural fault; NFT—north Fergana terrane; AIT—Atbashi–Inilchesk terrane; CCT—Central Caspian trough; NWSZ north-western side zone; NSZ —northern side zone; AASU—Astrakhan–Aktobe system of highs; SHE—South Emba uplift; ESZ—eastern side zone; TS—Tugarakchan sag; DT—Donbass–Tuarkyr inverted zone. Legend: 1—blocks with Precambrian continental crust; 2—Phanerozoic subduction–accretionary complexes; 3—strike–slip faults; 4—faults without separation; 5—state border of the Republic of Kazakhstan.

As a result of these events, a sedimentary sequence of the Upper Paleozoic, Mesozoic and, to a lesser extent, Cenozoic, was formed, composed of alternated siliciclastic–carbonate rocks separated by Kungurian salt with interlayers of anhydrites (Figure 2); however, in the northern part, it was separated with Roadian (Kazanian) salt, which makes it possible to divide the entire studied sedimentary sequence into subsalt and post-salt sedimentary complexes that are oil and gas bearing to varying degrees.

Salt-dome tectonics were widely manifested here, which led to the formation of more than 1500 salt domes with various shapes, sizes and heights of breakthrough of overlying strata and various combinations of structures in a plan view (Figure 3). Salt-dome tectonics determine the geological appearance of post-salt deposits, and are a characteristic feature of this depression [7].

The literature that has been devoted to the salt tectonics of the North Caspian basin is extensive, since the first hydrocarbon deposits here were confined to the arch parts of salt domes. However, the origin of the salt basin at the end of the Early Permian epoch has not yet been studied in detail. The formation of this basin took place during a change in tectonic epochs, when the Hercynian ended and the Kimmeridgian began. Then, the Proto-Tethys

ceased to exist and the Paleo-Tethys was formed, but the North Caspian basin was part of both. The salt basin in this area was preceded by a basin of carbonate sedimentation, when numerous isolated carbonate massifs were formed, widespread throughout Central Asia [8]. Salt accumulation in the second half of the Permian period was halted by the invasion of red-colored rocks, which also occurred over a wide area [9–11]. In this paper, we fill this gap in the research through our analysis of the reasons for the formation of the salt basin on the territory of the Caspian depression in the Kungurian time at the end of the Early Permian epoch.

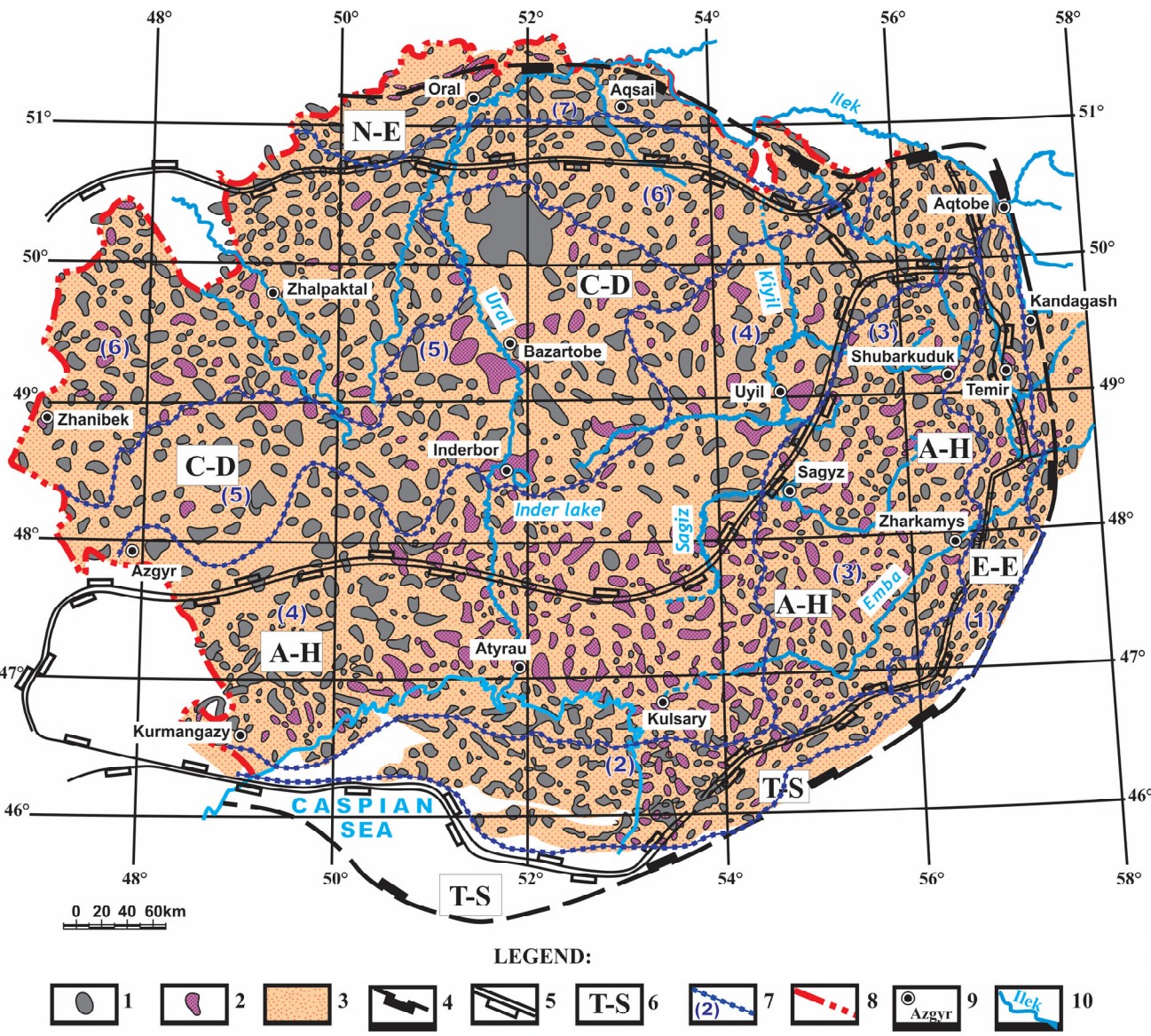

**Figure 2.** Distribution of salt domes in the North Caspian basin. Legend: 1—contours of salt domes identified from gravimetric and seismic data; 2—contours of salt domes, with identified oil and gas shows of overlying post-salt deposits; 3—inter-dome deposits of undivided post-salt Upper Permian–Triassic beds; 4—boundaries of the North Caspian basin; 5—boundaries of the main structural elements: 6—definitions: C-D—Central Caspian trough; A-H—Astrakhan–Aktobe system of uplifts; T-S—Tugarakchan sag; E-E—eastern edge zone; N-E—northern edge zone; 7—zones characterizing different types of salt diapirs: (1)—eastern; (2)—south-eastern; (3)—Uyil-Emba; (4)—Azgir-Khobda; (5)—central; (6)—north–north-western; (7)—northern; 7—letter designations of allocated zones; 8—settlements; 9—rivers; 10—state border of the Republic of Kazakhstan.

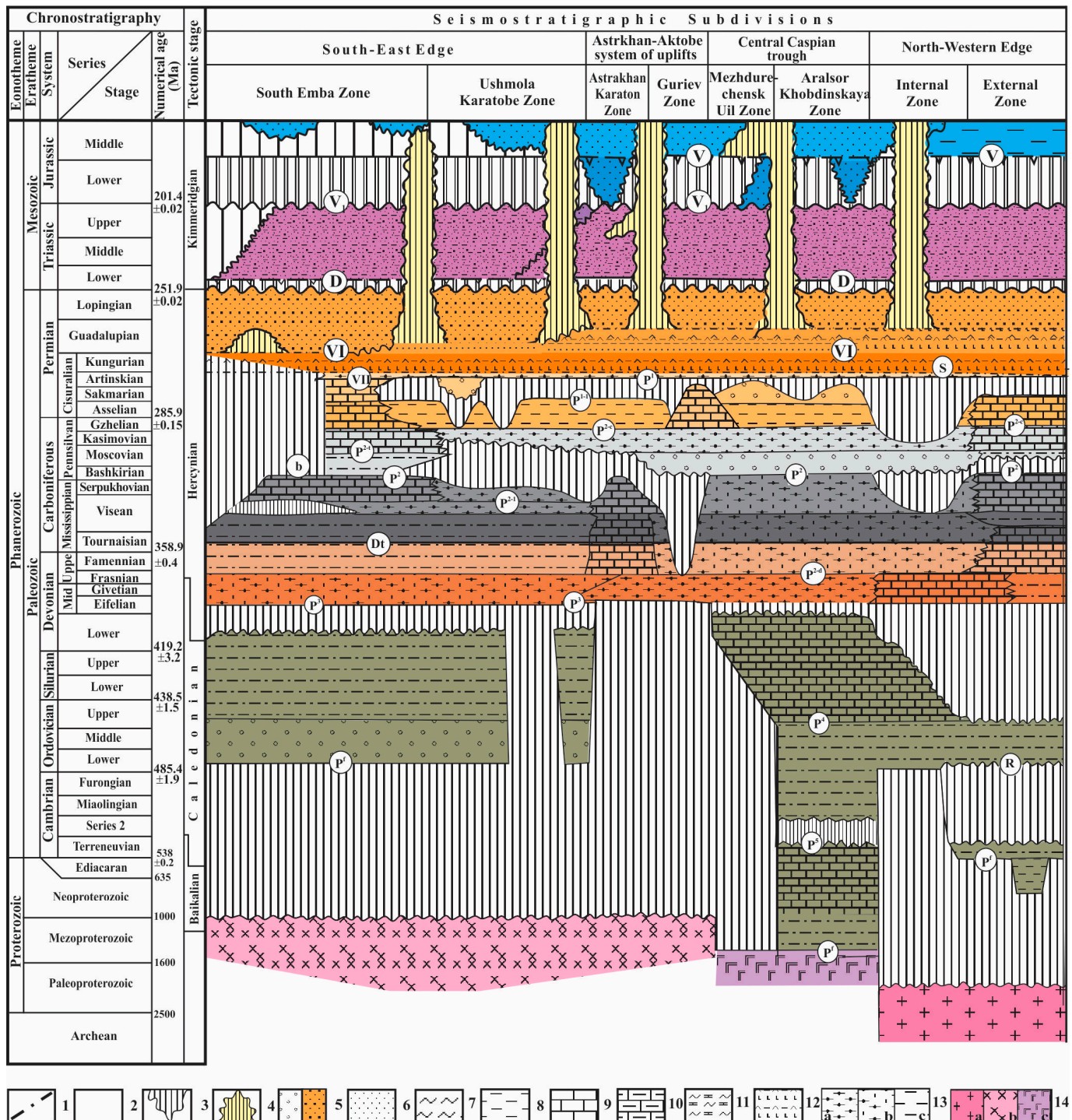

**Figure 3.** Seismic stratigraphy correlation chart (modified after [11]). Legend: 1—boundary; 2—sedimentary break and erosion; 3—sedimentary channel; 4—hiatus related to salt tectonics; 5—fluvial deposits; 6—shallow water marine siliciclastics; 7—shallow water marine siltstone; 8—argillaceous carbonate shales; 9—carbonates; 10—argillaceous carbonates; 11—cherts and shales; 12—salt and sulphate deposits; 13—Lower Paleozoic siliciclastics: a—coarse grain; b—sandstone; c—fine grains; 14—consolidated crust: a—granitic continental; b—oceanic basic; c—intermediate.

## 2. The Geological Features of the North Caspian Basin

The sedimentary succession of the North Caspian basin includes Neoproterozoic-alternated siliciclastic–carbonate strata, Ediacaran–Lower Paleozoic siliciclastic formations,

Upper Cambrian–Ordovician carbonate units, Ordovician–Silurian and Devonian–Lower Permian siliciclastic rocks associated with the carbonate massifs of the Famennian stage –Lower Carboniferous series, the halogenic strata of the Kungurian–Roadian age, the red-colored and variegated siliciclastic and terrestrial sedimentary units of the Upper Permian and Triassic period and the alternated siliciclastic–carbonate formations of the Jurassic, Cretaceous periods and Cenozoic era (Figures 3 and 4). According to the geological and geophysical data, the total thickness of the sedimentary cover in the center of the depression is estimated to be 20 km, decreasing to 10–12 km at the sides of the depression and to 3–5 km at the scarps. In the center of the depression, the approximate measurements of the different parts of the section are as follows: the Neoproterozoic is about 4 km, the Ediacaran–Lower Paleozoic 2 km, the Upper Ordovician–Silurian 2 km, the Devonian–Lower Permian 4 km, the Kungurian–Roadian part of the Section 4 km, the Upper Permian–Triassic 2 km and the Jurassic–Cenozoic 2.5 km [11].

In the North Caspian basin, along the surface of the basement (the consolidated Earth's crust), a number of large first-order tectonic elements are distinguished, including the northern side zone, the Central Caspian depression, the Astrakhan–Aktobe system of arched uplifts, the Tugarakchan sag and the eastern side zone (Figures 1 and 2). All of these played an important role in the formation of the basin at different stages of evolution. The sedimentary succession of the basin is traditionally divided into seismic reflective surfaces (Figure 3), which include the basement ($P^F$ reflecting horizon) and the main reference seismic horizons confined to the surfaces of structural unconformities; reflecting horizon $P^3$ is the surface of the pre-Devonian unconformity, reflecting horizon $P^1$ is the surface of the subsalt Upper Paleozoic unconformity of the end of the Early Permian epoch and reflecting horizon $V$ is the surface of the Pre-Jurassic unconformity. Sub-salt tectonic elements developed, which were inherited from the end of the Paleozoic era, including the Mesozoic and Cenozoic eras (Figure 4). The supra-salt elements experienced deformations associated with salt tectonics.

The deepest feature is the Central Caspian trough. It is contrastingly expressed in the structure of the mantle, basement and sedimentary cover. According to magnetic surveys, there is a clear negative regional anomaly, and in the gravity field this corresponds to high-intensity maximums of gravity anomalies. Anomalously high values of seismic velocities are characteristic of the rocks that make up the consolidated crust of the Central Caspian block. Boundary velocities vary from 6.7 to 7 km/s, and formation velocities in the upper part of the consolidated crust have consistent values of 6.5 km/s. Two seismic reflectors can be traced here, one at a depth of 32 km, the other at a depth of 42 km. In this case, the refracting horizon with Vr-8.0–8.1 km/s (Moho surface) coincides here with the uppermost areas of these boundaries, and the lower reflecting boundary is located at the level of the M surface outside the Central Caspian block [11,12].

Drilling materials on the eastern side of the North Caspian basin (eastern Akzhar, Kumsay and Baktygaryn locations) have shown the stratigraphic position of the reference seismic horizon $P^3$ [13]. This reflecting horizon has been studied only with the east Akzhar G-5 well, located on the Temir uplift, and two other wells, Baktygaryn G-1 and Kumsay G-4, in the same area, have revealed Devonian carbonate strata. The $P^3$ reflector is interpreted at the base of this sequence. The core samples studied from these deposits are represented mainly by carbonate rocks and contain numerous organic remains. The conodont *Ozarkodina remscheidenis remscheidenis* Ziegler was discovered there in core samples from the east Akzhar G-5 well, in the depth range of 5745–5751 m and 5738–5745 m [12], which indicate the lowest parts of the Lochkovian stage of the Lower Devonian, and clearly demonstrate that the reflecting horizon $P^3$ is confined to the base of the carbonate deposits of the Lower Devonian (Lower Lochkovian stage). Reflecting horizon $P^3$ is the reference horizon, and can be traced throughout the North Caspian basin. In the most submerged parts of the Central Caspian depression, the base of the lowermost Devonian deposits lies at a depth of about 14 km, and below it, between the basement surface and the reference

horizon $P^3$, there is a thick succession (up to 8 km) of siliciclastic and carbonate rocks, which has a pre-Devonian, late Neoproterozoic–early Paleozoic age (Figure 3).

The overlying section of the Devonian–Early Permian (pre-Kungurian) sedimentary sequences of the North Caspian basin is located in the interval between seismic horizons $P^3$ and $P^1$, and is subdivided into a number of seismic stratigraphic units (from bottom to top) (Figures 3 and 4): 1—Lower Devonian–Eifelian–Lower Frasnian from below, limited by the reflecting horizon $P^3$, and from above by the seismic horizon $P^{2d}$; 2—Upper Frasnian–Lower Visean, located between reflecting horizons $P^{2d}$ and $P^{2-1}$.

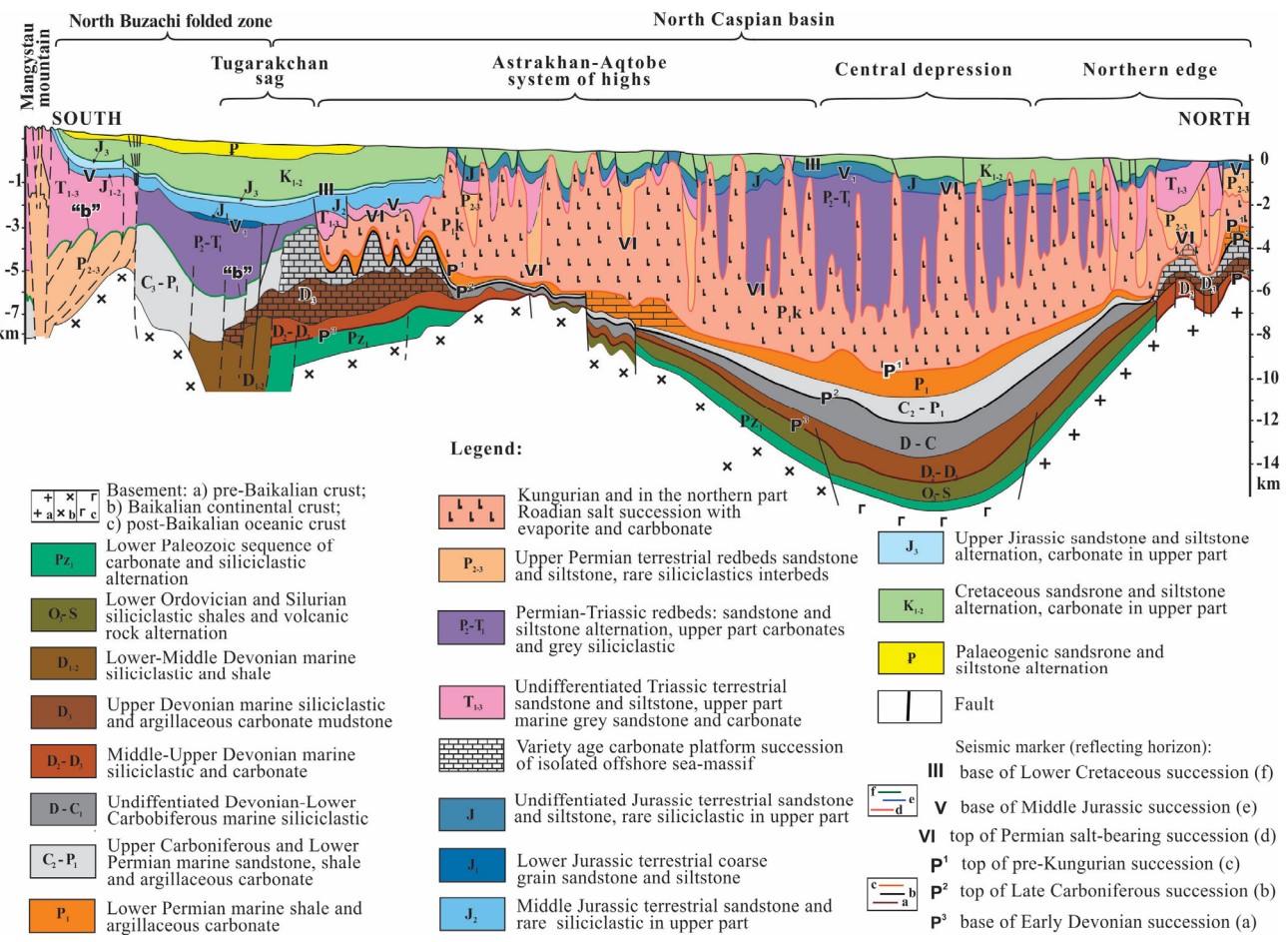

**Figure 4.** Generalized geological cross-section of the North Caspian basin from south to north (modified from [11,14]).

In recent years, studies in the southern part of the North Caspian depression have substantiated the identification of the seismic horizon $D_t$, which divides the complex under current consideration into two subcomplexes: Upper Frasnian–Lower Tournaisian and Upper Tournaisian–Lower Visean (on the northern side of the depression, the analogue of the $D_t$ horizon is the $C^1$ horizon); 3—Late Visean-Bashkirian ($P^{2-1}$ and $P^2$); 4—Moscovian–Lower Gzhelian, contained between the horizons from $P^2$ and $P^{2c}$; 5—Gzhelian-Early Permian, which is marked on top by the reflecting horizon ($P^1$).

In seismic stratigraphic successions, changes in the facies' composition are noted depending on the position of the territory within the zones corresponding to the regional geological (seismic–geological) zonation of the North Caspian basin.

Carbonate platforms of the Caspian basin are divided into two groups: epicontinental-attached carbonate platforms and intra-basinal off-shore submarine-carbonate island mounts. The first group includes the platform of the northwestern Caspian side and, apparently, the South Emba platform, limiting the depression from Northern Ustyurt. These platforms are associated with stable continental blocks. Isolated submarine-carbonate

platforms are located in the intracontinental deeper part of North Caspian, and were created by massive bodies (Figure 5). These carbonate strata are well-studied with seismic surveys and rare, deep wells [13–15]. They form locally isolated carbonate bodies, which on subsurface maps and seismic profiles have a positive aggradational geometry, protruding hundreds of meters and even several kilometers above the flat basin plain. It was found that carbonate massifs are characterized by an increase in the thickness of sediments compared to their simultaneous frame, and the narrow transition zone, where a change in thickness is noted, is called a carbonate ledge (slope). The carbonate massifs themselves are almost entirely composed of limestones and dolomites of different facies with rare interlayers of volcanic–siliciclastic and fine-grain siliciclastic material. Large well-known hydrocarbon deposits are associated with them, such as Kashagan, Tengiz, Karachaganak and Zhanazhol.

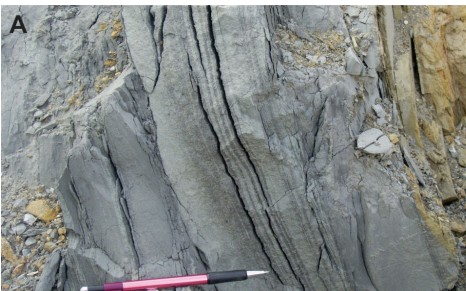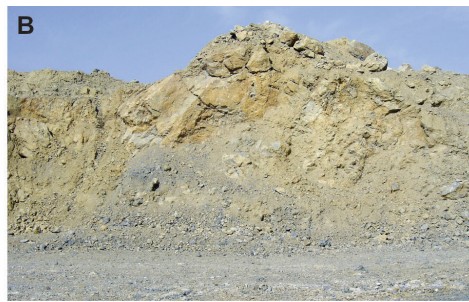

**Figure 5.** Layered (**A**) and massive (**B**) limestones of the Artinskian carbonate platform (mud mount), outcropping in the Aqtasty quarry in the vicinity of the Aqtobe city of the Aktyubinsk Cis-Urals.

Coeval sedimentary units developed along the framing of isolated carbonate massifs with a condensed (relatively thin) sequence are represented by dark-colored carbonate–clayey, mixed carbonate–siliciclastic and siliciclastic–shale, and, in more distant parts of the basin, thin-layered clayey–shale and siliciclastic sediments. Usually, such sedimentary beds are leveled and form a flat bottom of the basin, filling in any irregularities.

The age range of salt-bearing deposits has extended from the Latest Artinskian time of the end of the Early Permian to the Late Roadian (Kazanian) time at the beginning of the Late Permian [16]. The early assumptions about the widespread development of the Roadian (Kazanian) salt were not confirmed, and no biostratigraphically substantiated thick deposits of Roadian (Kazanian) youngest salt were found in the southern and eastern areas [17]. There, deposits of salt and evaporites are present, but are limited in the form of thin interbeds in red units. The main component of the salt-bearing formation is rock salt, containing interlayers of anhydrites, dolomites, potassium salts, siliciclastic and carbonate rocks (Figure 6). The structure of the salt-bearing deposits of both the Kungurian and Roadian sedimentary cycles is characterized by a clearly defined facies zoning, which consists in the fact that the relative total thickness of non-salt layers demonstrates many increases from the inner to the side parts of the depression. At the same time, their composition in the southeastern side zone is dominated by terrigenous rocks, and on the northwestern side, by sulfate–siliciclastic and carbonate rocks.

According to the seismic and gravity survey data and based on calculations, the primary sedimentary thickness of salt in the center of the depression could reach 4.5 km, and on the sides, 1–2.5 km [1,18]. The top of the salt area, which has a complex rugged topography due to salt domes and diapirs, forms a reflecting seismic horizon VI, and the amplitude of the salt domes and diapirs reaches several hundred and even thousands of meters.

Upper Permian continental redbeds, which lie above the salt-bearing Kungurian deposits at the base, are preceded by marine and lagoonal clayey-carbonate rocks, which are overlain by sandy–clayey red and variegated strata (Figure 7) [19].

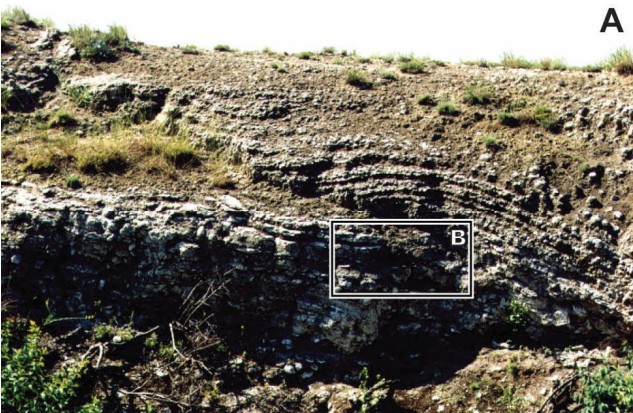
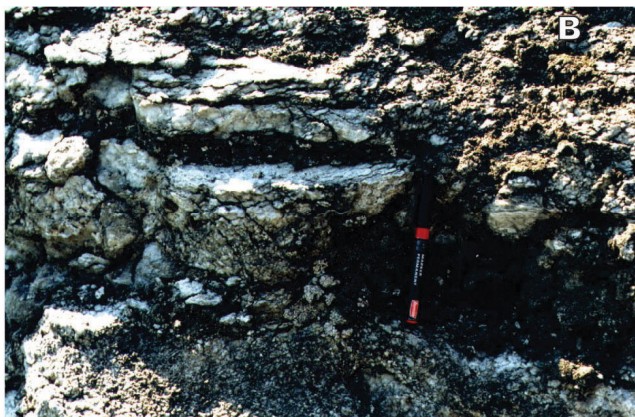

**Figure 6.** Outcrops of the Kungurian Formation (**A**) composed of rock salt (white), anhydrites (grey) and black mudstones. (**B**), the square with the letter "B" in the left figure indicates the location of the right figure, which shows more details. Fine parallel bedding is noted. Aleksandrov section, to the west, near the Aqtobe city, Aktyubinsk Cis-Urals.

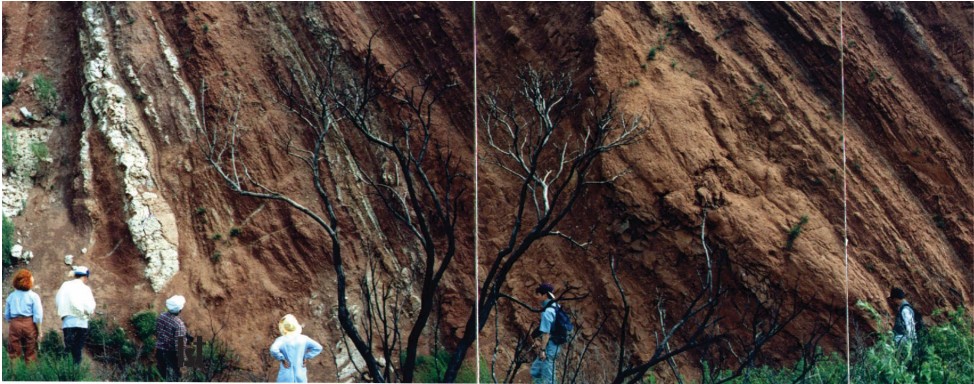

**Figure 7.** Interlayering of Upper Permian redbeds of polymictic sandstones and siltstones with interlayers of white limestones. Section along the valley of the Kurasha River to the west near the Aqtobe city, Aktyubinsk Cis-Urals.

These were mainly studied in wells and developed along the periphery of salt domes and in inter-dome depressions. The processes of salt dome tectonics completely disrupted the primary stratigraphic relationships of these strata, which complicates the development of stratigraphic schemes and, accordingly, creates difficulties in understanding the details of its structure. In the west of the depression below, limestones and dolomites predominate; in the east, there are mainly clays with interlayers of pink sandstones; and in the southeast, the section is dominated by red- and green-colored clays with interlayers of siltstones. Also, in the east of the depression in the upper part of the Upper Permian section, layers and a lens of conglomerates and brick-red sandstones are characteristic. These Upper Permian strata are replaced at the top by Triassic sediments, which are less deformed, but form a single sedimentary sequence with the underlying formations. In the west, the Triassic strata in the lower part are represented by reddish-brown clayey–silty deposits, which become coarser to the east and southeast and in areas where interbedded sandstones with interlayers of small-pebble conglomerates are noted. In the southwest of the Caspian region, there are carbonate clays. In the middle part of the Triassic section, closer to the side zones of the depression, sandy–clayey sediments have accumulated; in the southwest, they are replaced by gray-colored skeletal detrital and dolomitic carbonates, with interlayers of terrigenous material; in the southeast, clays with rare interlayers of carbonates predominate.

The upper part of the Triassic section Is eroded in many places, and where it exists, it is represented by clayey–silty deposits with sandstone interlayers with abundant fragments

of terrestrial plant detritus (Figure 8) [20]. These deposits are confined to the surface of a major regional unconformity characterized by seismic reflector **V**.

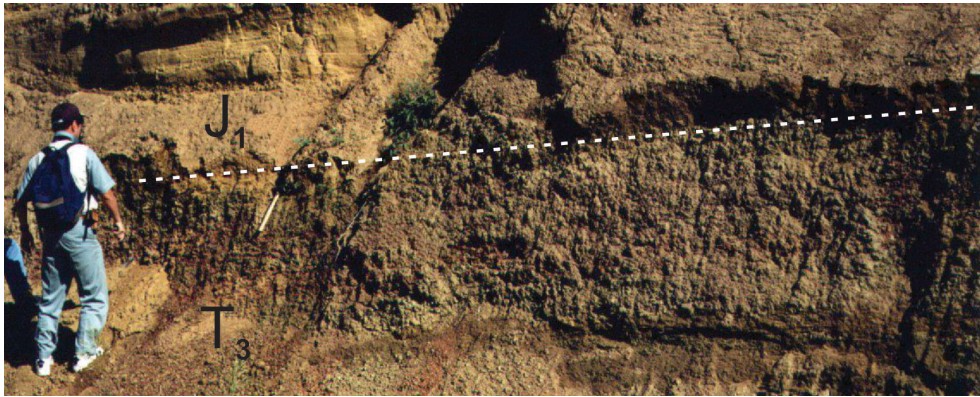

**Figure 8.** Exposed boundary of the Upper Triassic (greenish-red clay $T_3$) and Lower Jurassic (yellowish-gray sandstones $J_1$), with Dmitriev section to the west near the Aqtobe city, Aktyubinsk Cis-Urals.

## 3. The Regional Distribution of Carbonates

In our model, an important component is the Devonian–Carboniferous carbonates, formed as aggrading isolated off-shore island massifs. These played an important role in the final Late Paleozoic stage of the development of the North Caspian basin. Carbonates are important and significant features, and are characterized by a continuous sequence of deposits that make it possible to evaluate of the entire territory of depression.

The Devonian–Carboniferous carbonate sequences of the northern part of the North Caspian basin are known in two tectonic zones: the north–northwestern side and the Volgograd–Orenburg monocline system. Here, the carbonates form a continuous sequence from the Middle Devonian to the Early Permian and, at least, the Middle–Upper Devonian part of the sequence represents a single undifferentiated carbonate plate with intercalations of siliciclastic material; meanwhile, for the overlying part of the Upper Devonian–Carboniferous–Lower Permian deposits, a separate Karachaganak carbonate massif is identified [11,13,15].

The carbonate deposits of the southern and southeastern parts of the depression are limited to the zone of the Astrakhan–Aktobe system of uplifts, and form a complex, isolated Tengiz–Kashagan carbonate massif, varying in age from the upper Devonian (Frasnian–Famennian) to the Lower Middle Carboniferous (Bashkirian) (Figures 9 and 10).

In the area of development of these carbonate deposits, the Devonian (Late Frasnian–Famennian) part of the section, in places, forms a carbonate plate of a significant area, and the differentiation of carbonate massifs is more typical for the carbonate part of the section. At the junction of this zone with Northern Ustyurt, the Yuzhnyi carbonate massif, which has a complex structure, has been identified. Carbonate deposits in the eastern part of the North Caspian region are distributed in two different geological settings. The first one corresponds to the Temir carbonate massif, located in the zone of the Astrakhan–Aktobe uplifts and consisting of a continuous sequence of, possibly, Devonian and Middle Carboniferous carbonates, surrounded by coeval depression sediments (Figures 9 and 11). The second one is represented by carbonate and alternated carbonate–siliciclastic deposits of the Serpukhovian–Early Permian age, making up the structure of the South Emba uplift, as well as extending north into the region of the Zharkamys uplift and its eastern framing. The structural features of the South Emba structure are expressed by a large antiform (anticlinal) fold of eastern vergence, with a gentle eastern wing and a steep western wing, complicated by steeply inclined faults. The core of the antiform, the so-called Mynsualmas ledge (Figure 11 (j)), is composed of siliciclastic sediments of the Upper Devonian–Early Carboniferous, and the limbs are composed of Upper Visean–Lower Permian carbonates. On seismic sections and on geological and geophysical profiles it is clearly noticeable that the central part (limb) of the antiform is eroded and unconformably overlain by the Middle Jurassic peneplain surface (seismic-reflecting horizon V). The level of erosion in the central part of the structure reaches the bottom of the terrigenous section.

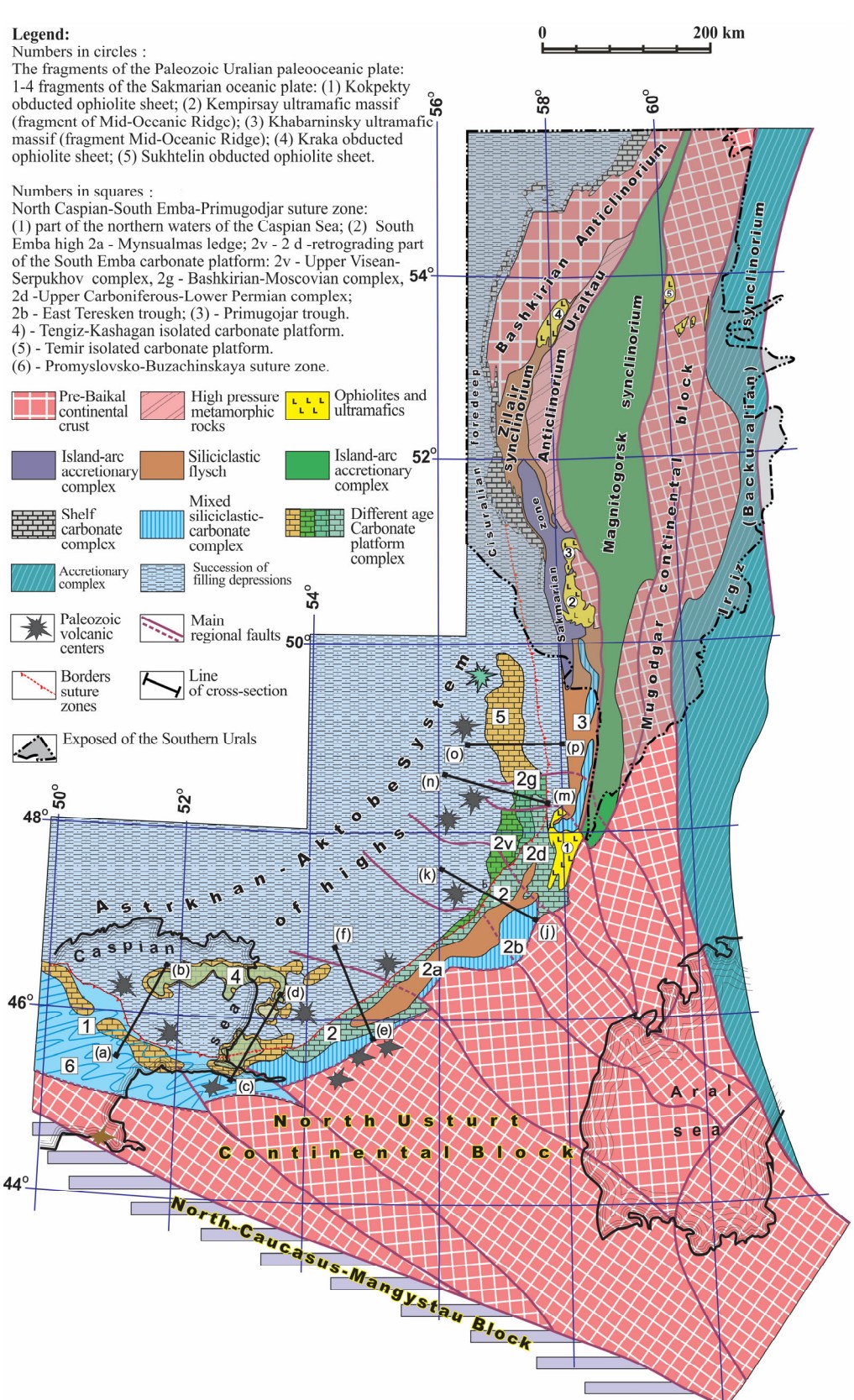

**Figure 9.** Scheme of the location of main geodynamic elements of the eastern margin of North Caspian depression, attached to North Ustyurt block and Uralian fold belt (modified from [13]).

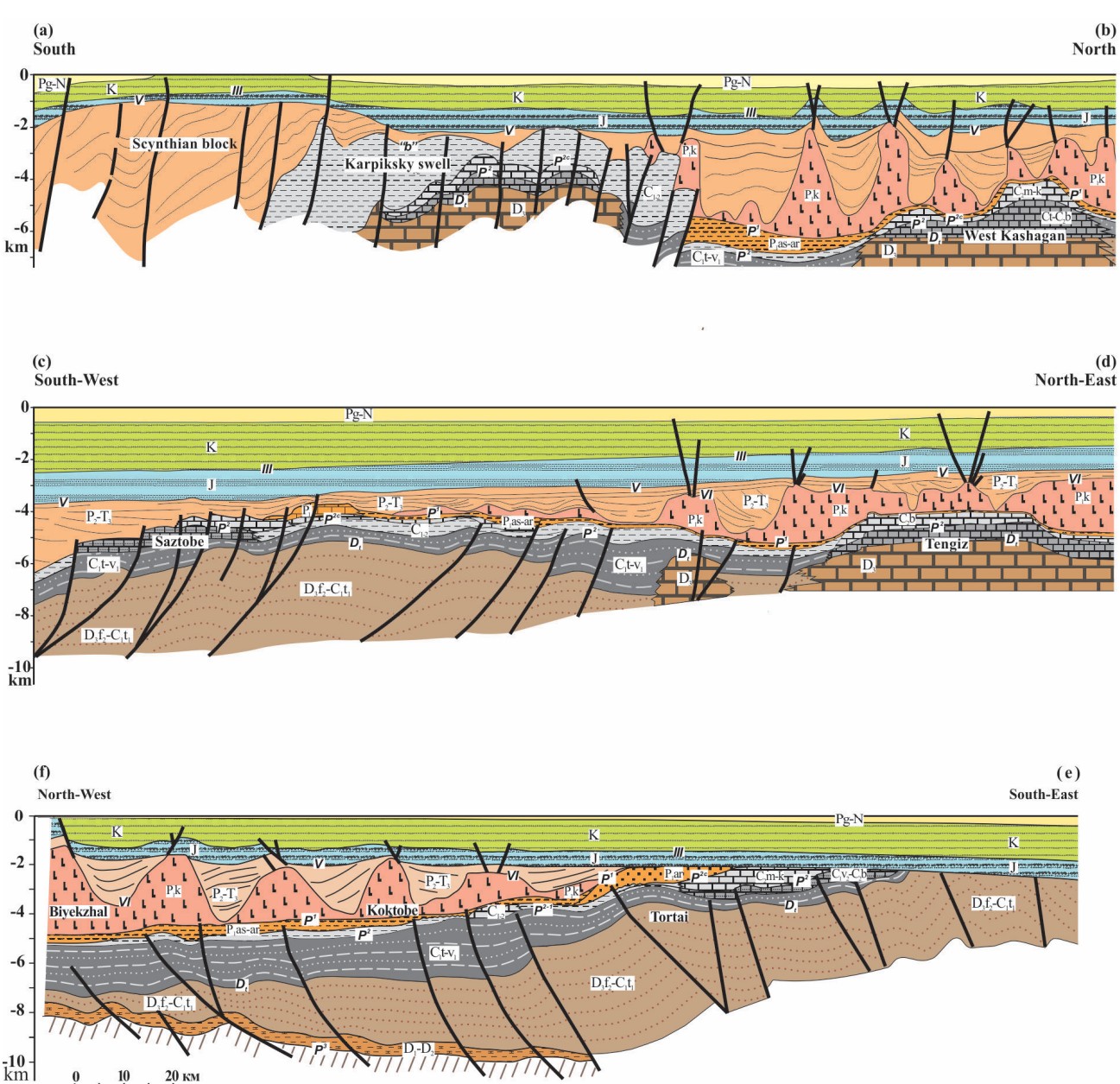

**Figure 10.** Geological cross-sections of southern part of North Caspian basin. For location, see Figure 5. **Line a–b**—in the direction from Scynthian block to western Kashagan carbonate platform. **Line c–d**—in the direction from Saztobe carbonate platform to Tengiz carbonate platform. **Line e–f**—in the direction from Biyekzhal salt dome to Tortai antiform. For the legend, see Figure 7 (modified from [13,15]).

The western limb of the structure, the part adjacent to the castle, is also eroded. As it moves west towards the central part of the North Caspian depression, here Middle Carboniferous (Moscovian–Kasimovian) carbonates and Lower Permian Artinsky–Sakmarian carbonates are brought to the level of the Middle Jurassic peneplain surface. The eastern part of the structure is completely buried under a thick succession of Permian–Triassic siliciclastic sediments and forms the so-called East Teresken trough. To the south, the degree of deformation of the South Emba antiform decreases, and periclinal closure is presented there.

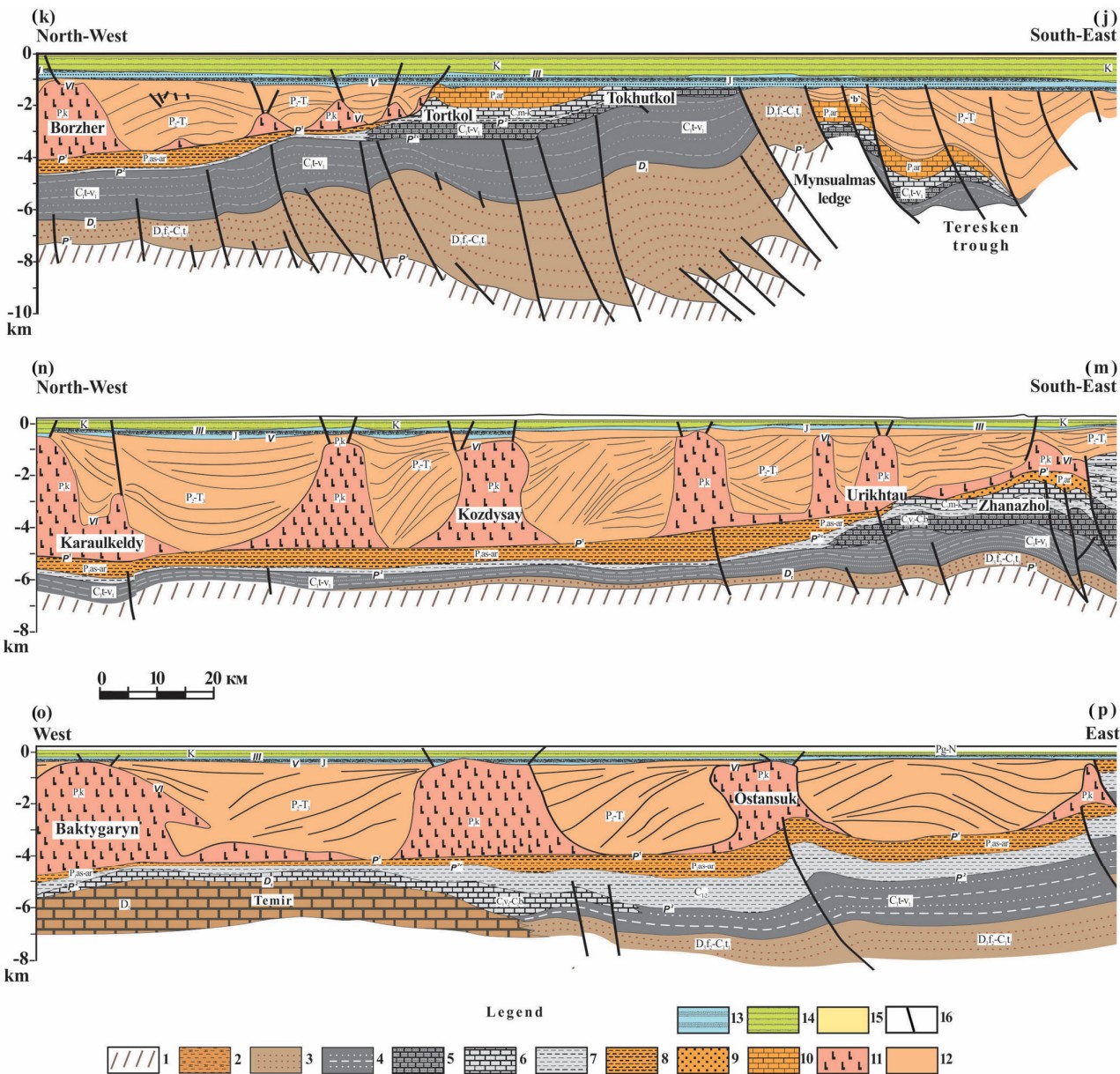

**Figure 11.** Geological cross-sections of eastern part of North Caspian basin (modified after [13,15]), location see in Figure 5. **Line k–j**—in the direction from Borzher salt dome to Tortkol–Tokhutkol carbonate platform. **Line n–m**—in the direction from Karaulkeldy salt dome to Zhanazhol carbonate platform. **Line o–p**—in the direction from Baktygaryn salt dome to Ostansuk salt dome (modified after [13,15]). Legend: 1—pre-Hercynian basement; 2—Early Devonian formation of gray siltstone, argillaceous shales and rare sandstone; 3—Upper Devonian green-graded siliciclastics; 4—Lower Carboniferous green-graded siliciclastics; 5—Lower Carboniferous carbonate platform interior sequences; 6—Upper Carboniferous carbonate platform interior sequence; 7—Carboniferous deep water shales and mudstone, alternation siliciclastic and carbonate formation; 8—Lowermost Permian siliciclastics and shales; 9—coarse-grain Lower Permian formations; 10—Lower Permian carbonate platform succession; 11—Kungurian salt-bearing units; 12—Middle and Upper Permian redbeds; 13—undifferentiated Jurassic units; 14—undifferentiated Cretaceous units; 14—undifferentiated Tertiary units; 16—faults.

Apparently, the carbonate deposits of the South Emba uplift represent a relatively young carbonate platform, deformed in Late Permian and/or Triassic times. The South Emba antiform occupies the North Caspian–South Emba–Ural zone, and its western

part overlaps the Astrakhan–Aktobe system of uplifts. Carbonate strata of the Lower Carboniferous–Lower Permian extend beyond the boundaries of the South Emba uplift in the north, and, in the continuation of this zone in the southwest–west direction, deformed carbonate deposits are conventionally identified, which are compared with the carbonates of the South Emba. We will call this entire carbonate complex the South Emba–Zhanazhol carbonate platform.

In some places, depressional Lower Permian strata are complicated by protruding carbonate massifs; for example, in the Tasym carbonate platform of the early Permian [14,21], which was discovered on the northern slope of the Astrakhan–Aktobe uplift. These are carbonate strata; the part uncovered by the well reaches a thickness of about 500 m and correlates with the Carboniferous–Early Permian carbonates of the South Emba, studied along the Tortkol–Tokhutkol line [13,22], as well as with the Permian carbonates of the Karachaganak platform and the northern marginal zone [11,23].

## 4. The Formation of the Carbonate Platform in the North Caspian Basin

The North Caspian salt-bearing basin is an independent structure formed only at the end of the Early Permian, when orogenic events occurred along its eastern and southern sites, in the southern Urals and in the south and southwest in the Donbass–Tuarkyr rift system. From that instant, the depression began to develop in its modern contours, whilst before, different parts of the Caspian depression belonged to different tectonic domains.

The early history of this depression is only now being reconstructed in general terms, and on the basis of comparisons with the framing regions of the East European platform and the Central Asian orogenic belt, which include the ancient continental blocks of Kazakhstan and the Tien Shan [3,6,24–29]. In the Middle Proterozoic era, this territory was part of the southern passive margin of the ancient East European continent. Here, at that time, there was a shelf basin open to the east towards a deep oceanic basin called the Proto-Ural basin [30]. During the collision of the continental masses of Baltic and Gondwana and the formation of the Rodinia supercontinent during the Proterozoic eon, the ocean disappeared, and in its place the Cadomian orogenic belt formed, a fragment of which now represents the complex structure of the Astrakhan–Aktobe block. Before its front, at the end of the Neoproterozoic, a marginal trough formed in the territory of the Central Caspian depression [8]. This scenario appears to be plausible, and can be compared with a recently developed model for the ancient geological history of the Karatau Range of southern Kazakhstan and the Middle Tien Shan, where close tectonic events at the end of the Neoproterozoic era led to the unification of the Central Kazakhstan and Tien Shan continental blocks with the supercontinent Rodinia [31].

From the very end of the Neoproterozoic era in the Cryogenian and Ediacaran periods, the orogenic structures of the Cadomian epoch underwent destruction. The rifting of this time led to the formation of a rift system, from which the Uralian rift basins were subsequently formed in the east, and the Tugarakchan rift basins in the south. The Neoproterozoic–Ediacaran stage of development of the region was one of decisive importance in the formation of the consolidated crust of the Central Caspian block. At that time, the main distinctive features were laid down in the structure of the earth's crust and in the upper mantle, which later had a decisive influence on the Paleozoic history of the development of the Caspian basin. In the Central Caspian depression, a lens of eclogites protruded, consisting of alternating mantle and oceanic ultramafic and mafic rocks [12]. The protrusion of a lens of such rocks with significant anomalous density disrupted the isostatic equilibrium, and ensured the gradual subsidence of the basement of North Caspian basin over the next almost 450 million years, until the end of the Permian period.

At the end of the Early Paleozoic era, in connection with the opening of the Ural Ocean and the ongoing rifting in the Tugarakchan trough, the Astrakhan–Aktobe system of uplifts, which bounded the Central Caspian depression from the south, was involved in the subsidence, and the epicontinental basin connected with the Proto-Tethys Ocean adjacent to the east. The Late Ordovician–Silurian period is marked by the fact that the

entire territory of the Caspian depression was part of a vast single basin located along the southeastern passive margin of the east European continent. Then, on the periphery of the Proto-Tethys, collision processes began, and this led to the formation of a folded belt on the southern frame of the North Caspian basin. At the same time, a collision and unification of the Astrakhan–Aktobe and North Ustyurt tectonic blocks may have occurred.

In the Early Devonian epoch, on the border of the Caspian depression and the North Ustyurt block, which was part of the Caledonian orogenic structure, the marginal Tugarakchan trough formed and created the conditions for a basin not compensated by sedimentation. In the resulting local topodepressions, a large amount of terrigenous turbidity was captured, carried down from the surrounding uplifts. Since the Late Devonian, carbonate platforms began to form here [13]. Their emergence was extended over time, which can be confidently confirmed by the different ages of their foundations. If we take into account that the Tengiz–Kashagan, Temir and Karachaganak carbonate massifs are isolated underwater carbonate mountains, and that they are located on a heterogeneous basin basement with a Lower Paleozoic structural stage and vary in thickness in Lower–Middle Devonian deposits, then we can talk about the initial Hercynian transgression on the dissected relief of the basin.

The heterogeneous basement was a consequence of the dissected topography of the basin, formed during the Caledonian stage of tectogenesis. Evidence of the Caledonian orogenesis can be provided by dacite volcanics exposed under Lower–Middle Devonian carbonates in the East Akzhar G-5 well [13] and, possibly, andesitic volcanic-exposed wells on the Buzachi Peninsula [32], in the southern frame of the Caspian depression. According to the geophysical data, the volcanogenic–magmatic complexes are widely interpreted there [33]. This scenario for the formation of carbonate massifs assumes that their bases are of different ages, since as the transgression developed, the relief uplifts were flooded one by one and a space was formed between the bottom and the sea level, which contributed to the upward growth of the carbonate massifs. During the initial stages of flooding, depressional deep-sea facies also accumulated in isolated basins. The slow nature of the initial transgression [34] led to the formation of extensive carbonate platforms, and the acceleration of marine flooding led to their differentiation and division into isolated carbonate massifs, as indicated by their cone-shaped shape. Along the framing of carbonate massifs and carbonate platforms, a deep-sea carbonate–shale depressional complex continued to form.

In the middle-to-late Devonian epochs, and until the middle of the Early Carboniferous epochs in the east and south-west of the Caspian Sea, there simultaneously existed a carbonate sedimentation basin, where carbonate massifs were formed, and a siliciclastic sedimentation basin, where the siliciclastic flysch analog of the Izembet (Zilair) strata accumulated. The introduction of a large amount of siliciclastic turbidity into the basin should have stopped carbonate accumulation; however, it must be assumed that there was a catcher between the two basins that prevented the penetration of terrigenous material to the west-southwest from the eastern areas of erosion. Such a barrier could be the Tugarakchan trough.

By the end of Visean time, all uplifts in western Kazakhstan were denuded and tectonic stabilization began. It is known that the end of the Visean time in Central Asia was marked by a maximum transgression, when the sea blocked almost all known land uplifts and carbonate deposits became widespread, and it was this transgression that led to the emergence of the South Emba–Zhanazhol carbonate platform in the Caspian basin.

The difference in the structure of the carbonate strata of the north–northwestern ledge, and the morphology of the South Emba–Zhanazhol carbonate platform, can be explained by the major restructuring of the sedimentary basins of Central Asia and its influence on the basins of western Kazakhstan. In post-Bashkir times, a collision of continental blocks occurred here, which led to a large left-sided shift of all tectonic blocks relative to each other [6,35]. Over a vast area, tectonic blocks began to move laterally along a system of curvilinear faults, which contributed to the simultaneous formation of zones of

local extension and zones of local compression on a regional scale. For the Ural orogen in the area of the Sysert–Ilmenogorsk zone and eastern Mugodzhary, it is assumed that the formation of large compression structures dates back to pre-Kungurian time. Deformations similar in age, which led to the cessation of carbonate accumulation in the south of central Kazakhstan and the southern Tien Shan, are estimated to be of post-Bashkir time, with a maximum stage for their manifestation in the Permian period [6,35,36].

## 5. The Geodynamics of the Salt Accumulation in the North Caspian Basin

The proposed geodynamic model is based on the assumption that the formation of the Caspian depression occurred at the end of the Proterozoic eon, when the Rodinia paleocontinent broke up; however, in the Paleozoic era, as a result of rifting, the Central Caspian depression was formed, filled with a thick multi-kilometer alternated siliciclastic–carbonate succession. From the east–south–east, it was terminated by the Aktobe–Astrakhan tectonic block. Further to the east, they were framed by an oceanic rift, which had a continuation and connected with the Sakmara oceanic rift of the western Urals. Even further east, the entire triad was framed by the continental massif of northern Ustyurt.

The North Ustyurt continental massif began to move closer to the Caspian basin, which was part of Eastern Europe as a continental margin, at the end of the Early Paleozoic, and this process was accompanied by frontal compression and the western (in modern coordinates) subduction of the oceanic plate of the Sakmara rift under Eastern Europe (Astrakhan–Aktobe zone of the Caspian basin). As a result of the Caledonian orogeny, a dissected relief was formed, and the Trans–Volga–Ural suture-folded zone stretched from northern Buzachi, through Southern Emba, and came to the surface in the Sakmara zone of the western Urals (Figure 9).

This is confirmed by the fact that the Izembet (Zilair) flysch sequence in a stripe frames both the Sakmara zone of the Urals and the Astrakhan–Aktobe block, as well as by the fact that the dacites discovered by the East Akzhar G-5 well and the supposed Paleozoic volcanogenic strata of the eastern margin of the Caspian Sea [33] can be correlated with island-arc volcanics of the andesite–dacite–liparite formation of the Baiterek Formation at the junction of the Sakmara zone of the Urals and the eastern margin of the East European Platform. On the other hand, the latest interpretations of the Ural fold belt [37,38] show that during the closure of the Magnitogorsk synclinorium, subduction was directed to the east and either a change in the polarity of subduction occurred or two-way subduction occurred.

This mechanism is a tool that explains the asymmetry of the zone of the Tugarakchan trough and the formation of a folded suture on the border of the Caspian Sea with Northern Ustyurt. With such paleotectonic reconstructions, it is possible to explain the appearance in the east of the Caspian Sea at the end of the Devonian period and the beginning of the Carboniferous period of huge areas of erosion in the form of island arcs and continental massifs and the clinoform morphology of the Izembet (Zilair) flysch layers.

The lithological composition of flysch deposits varies in strike from greywackes (products of the erosion of island arcs and oceanic complexes) [33] to lithite and arenite wackes, containing abundant fragments of the erosion of the ancient continental basement. The absence of clearly defined island arcs and their lateral attenuation along strikes [26] can be explained by the fact that, from the end of the Devonian period to the end of the Paleozoic era, the Ural suture zone was formed under conditions of the left-sided displacement of tectonic plates (see the discussion on Late Paleozoic strike–slip deformations in Central Asia in [6,26,35,37].

A change in the polarity of subduction and left-sided shear deformations along the curvilinear route of the Main Ural Fault, as well as along its possible continuation along the northwestern frame of the Northern Ustyurt block, could lead to the fact that a part of continental plate of the North Caspian basin was pulled under the continental massif; that is, "Ampferer" subduction occurred, which was synchronous with the normal Benioff subduction in the Magnitogorsk (Greenstone) zone of the Urals. This process was accompanied by the compression and formation of an orogen in the southern Urals and

subsidence within the east North Caspian basin. At the final stage of subduction, when all the oceanic plates were absorbed, suture (accretionary-island-arc) complexes were formed, rigid continental plates were brought into contact and the deformation (or inversion) of the flysch clinoform occurred, which formed the foundation for the accumulation of carbonates on the South Emba (Figures 12 and 13). This scenario of geodynamic evolution significantly simplifies the sequence of geological events, and better links the geological structure of the Ural fold belt and the Caspian basin. Frontal western subduction and oblique (left-sided) eastern subduction explain the differentiation in the thickness of Paleozoic deposits in different zones of the Caspian basin and the difference in the structure of carbonate massifs and carbonate platforms. Early Caledonian tectogenesis formed the Aktobe–Astrakhan uplift system as the margin of the East European Platform, with a dissected relief and sedimentation barrier in the form of the Tugarakchan trough, and the complex kinematics of tectonic blocks led to its long-term subsidence and the formation of carbonate massifs, which were isolated as submarine carbonate mountains (platforms).

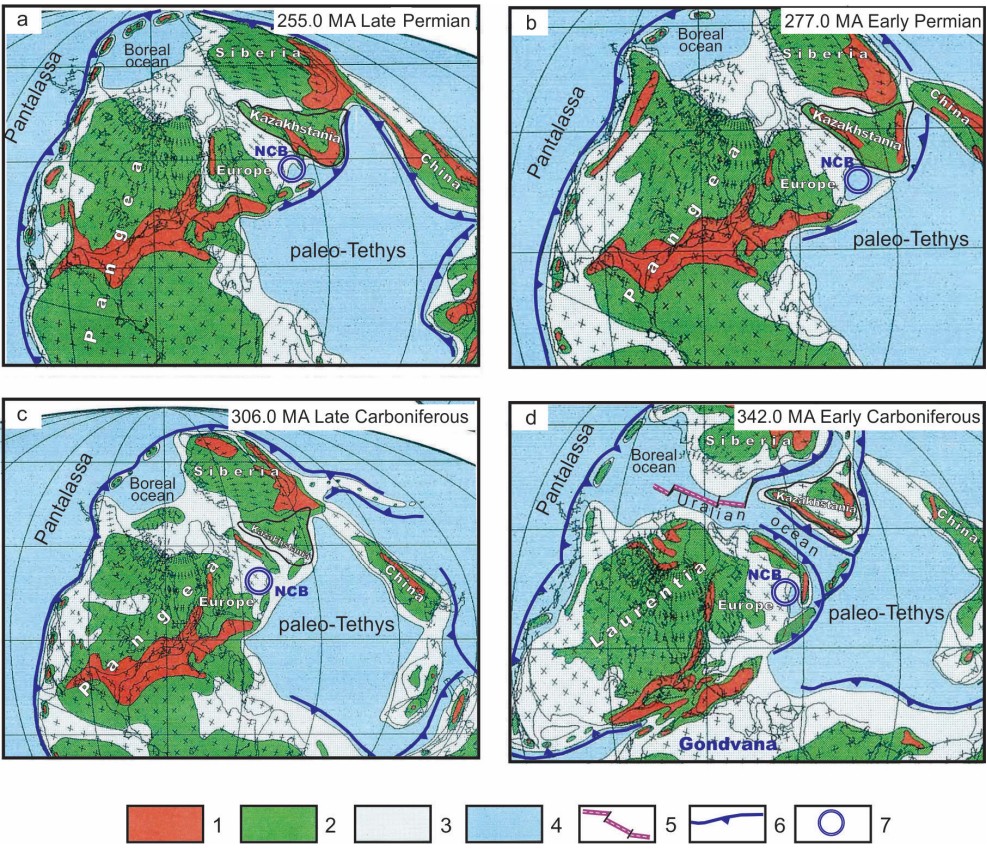

**Figure 12.** Paleotectonic reconstructions of Eurasia and neighboring continents through (r2) geologic time (modified after [4]); (**a**) 255 million years ago, the Ural orogeny (mountain building) and the invasion of redcolored sediments into the North Caspian basin, (**b**) 277 million years ago, active collision of ancient continental blocks within the continent of Pangea and restructuring of the northern margin of the Paleo-Tethys, (**c**) 306 million years ago, the beginning of the unification of continental blocks into a single supercontinent Pangea, final formation of the North Caspian basin, (**d**) 342 million years ago, closure of the Ural Ocean and separation of the Caspian basin. Legend: 1—mountains; 2—landmass; 3—shallow marine and/or submersed continental shelf and margin; 4—deep water; 5—mid-oceanic ridge; 6—trench and subduction; 7—North Caspian basin (NCB).

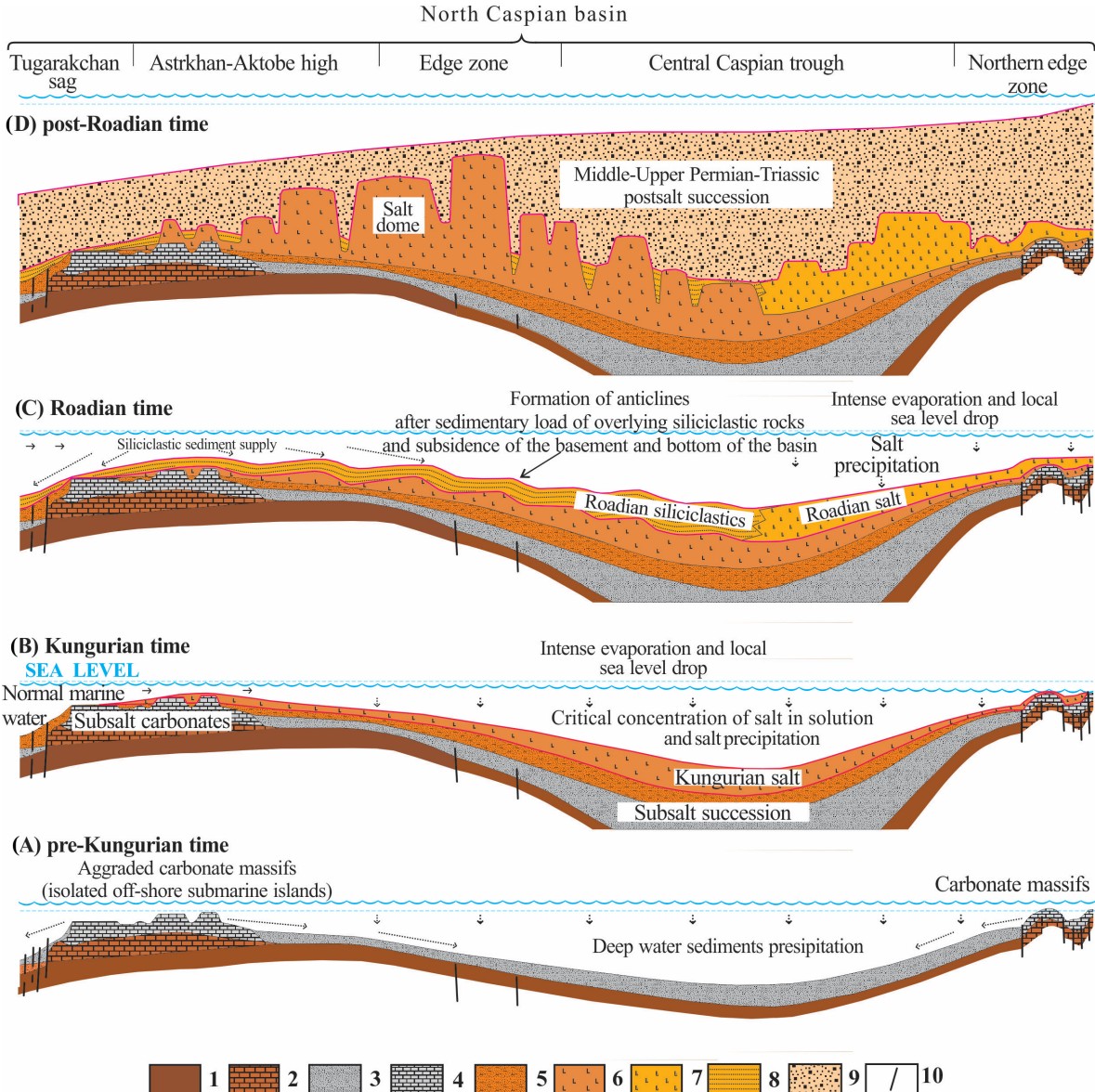

**Figure 13.** Diagram of the formation of the salt-bearing Caspian depression, from the stage of salt accumulation in the Kungurian time to the stage of salt diapirism (modified after [7]), (**A**) pre-Kungurian time, the formation of the North Caspian basin and along its framing the growth of island uplifts composed of carbonate massifs, (**B**) Kungurian time, isolation of the North Caspian basin from the waters of the Tethys Ocean and, due to intense evaporation and unidirectional influx of sea water, deposition of rock salt, (**C**) Roadian time, in the south, the invasion of redcolored sediments has begun, and in the north, salt accumulation continues, (**D**) post-Roadian time, very large invasion of redcolored sediments and the beginning of salt dome tectonics. Legend: 1–Devonian subsalt sediments, predominantly of mixed siliciclastic-carbonate composition; 2—Upper Devonian carbonate deposits; 3—mixed siliciclastic–carbonate deposits of the Carboniferous; 4—carbonate deposits of the Carboniferous; 5—siliciclastic–carbonate deposits of the Lower Permian; 6—salt and sulfates of the Kungurian stage of the upper Lower Permian; 7—salt and sulfates of the Roadian stage of the lower Upper Permian; 8—siliciclastic deposits of the Roadian stage of the lower Upper Permian; 9—siliciclastic sandstones and siltstones of the Upper Permian and Triassic; 10—tectonic faults.

The final stage of the collision of the Caspian region and northern Ustyurt is associated with the Hercynian deformations and stabilization of the tectonic regime in the middle of the Early Carboniferous epoch and the formation of the South Emba–Zhanazhol carbonate

platform, which, unlike the Tengiz–Kashagan and Temir carbonate massifs, was formed as a carbonate platform framing the stable continental block of northern Ustyurt.

Deformations from left-sided to right-sided, which occurred in Central Asia at the end of the Permian period [6]. Taking into account the curvilinear trajectory of the faults, it can be assumed that the eastern vergence of the South Emba antiform and the western vergence of the folding of the Cis-Ural foredeep are the result of dextral shear events along regional faults of the northwestern strike. That is, during the regional right shift at the end of the Permian period, the eastern margin of the Caspian region experienced compression in the direction of Northern Ustyurt, which caused the breakdown of the carbonate strata, its deformation and erosion, as well as the formation of the East Teresken trough.

The deformations of the South Emba carbonate platform were apparently associated with transpressive (convergent) strike–slip inversion, or a change in the sign of strike–slip.

For the geotectonic model, the assumption that explains the subsidence of the Earth's crust in the North Caspian basin under compressive tangential forces is worthy of attention, and additionally suggests that, since at the base of its crust there is an eclogite lens, it is the force source responsible for the blocks' subsidence. [12]. This subsidence mechanism, associated with island uplifts along the periphery of the basin, caused the uplift of the carbonate massifs, thus forming a sedimentation barrier that contributed to the accumulation of Caspian salts at the end of the Late Permian epoch in the Kungurian time, and to the north during the Roadian time of the Early Permian epoch [12,26,27].

In pre-Kungurian times, the carbonate massifs were flooded when the depth of the sea above them was so great that the deposition of carbonate sediments ceased. If we combine the facts, and, in particular, the regional geological models, then the increase in the basin depth is more easily explained by the flooding of the carbonate massifs. The continuous sequence of Serpukhovian–Lower Permian carbonates on the South Emba and the introduction of thin siliciclastic units with the simultaneous retrogradation of the South Emba carbonate platform to the east also confirm this phenomenon. The deepening of the Caspian basin in post-Bashkir times was most likely not due to eustasy, but to the subsidence of the foundation of the Caspian basin during the movement of tectonic blocks along curvilinear trajectories. This event was preceded by the accumulation of fine-grained carbonaceous depressional carbonate–clayey-siliciclastic sediments covering the entire basin [37], as well as the overlying Lower Permian sediments. With such phenomena, the subsidence of the bottom of the basin is also consistent with the presence of eclogite lenses, which drag it into subsidence.

The nature of the Kungurian salt accumulation in the North Caspian basin is compared with the modern Karabogaz–Gol Bay area, when, as a result of intense heating of sea waters in a wide yet restricted water area, salt accumulation occurs due to intense evaporation [39]. In such a model, for a salt basin, it is important to have a physical barrier in the form of a local uplift that can change the depth of the basin bottom. It interferes with normal water exchange with the open ocean. As a result, the concentrations of dissolved salts in the basin do not equalize. The long-existing arid paleogeographical environment contributed to the active evaporation of sea water in the basin, a decrease in the local sea level and, as a consequence, allowed for a constant flow of sea water [40] from the Paleo-Tethys. All these events led to the accumulation of significant thicknesses of salt-bearing deposits (Figure 13).

The sedimentary barrier could have been provided by numerous and different-aged carbonate massifs protruding above the bottom, located in the east and south, disconnecting it from the Paleo-Tethys and limiting water exchange between the depression with an abnormally high salinity of water and the open ocean possessing normal marine salinity. The massifs were associated with a previous epoch of sedimentation, were not compensated for by low sedimentation rates and did not level the basin floor. The formation of uplifts due to island land is not assumed, since there is no evidence of the influx of large volumes of terrigenous turbidity that would interlayer the salt-bearing layers. Such island uplifts formed around the Caspian basin much later, in the Guadalupian era, and did not form simultaneously. In the very south, this happened earlier, since the salt basin limited its

existence at the beginning of the Guadalupian epoch. In the northern part of the depression, salt accumulation continued, and was limited only in the middle of the Guadalupian epoch for the same reason as in the south, due to the introduction of siliciclastic turbidity from the island land, and with it the presence of fresh water, which reduced the abnormal stability of salt in the water and stopped its accumulation at the bottom of the basin.

The end of the Carboniferous period and the beginning of the Permian period for the Caspian basin is the time when events occurred along its eastern and southern framing that led in the east to the collision of large continental blocks of Kazakhstan and Siberia with the East European Platform [3,27] (Figure 12). These collisions caused the invasion of a large volume of red-colored strata at the end of the Permian period, filling both the Caspian depression and the Cis-Ural marginal trough, and in the south the territory of Mangyshlak and the Karpinsky swell. For the Caspian basin, this mechanism of subsidence of the basin basement adequately explains the formation of many kilometers of red-colored coarse-grained strata (Figure 13), the accumulation of which had already occurred there in the second half of the Permian period, when salt accumulation ceased in the post-Roadian time. With the accumulation of thick Upper Permian–Triassic terrigenous redbeds, salt tectonics also started and formed salt domes.

## 6. Conclusions

In the geodynamic sequence of events, a number of characteristic features can be identified that changed the basins of western Kazakhstan and moved them in the direction of their tectonic stabilization.

(1) In the Precambrian–Early Paleozoic era, the Caspian basin formed as a continental rift structure with a maximally sagging central part. In the east, it was limited by an oceanic rift, which separated the eastern margin of the east European craton (Astrakhan–Aktobe system of uplifts) and the continental block of northern Ustyurt. The depression was filled with sedimentary strata of significant thickness. At the end of the Early Paleozoic (Caledonian tectogenesis) era, the continental blocks of the North Caspian basin and northern Ustyurt began to converge, as a result of which a dissected relief was formed and the closure of the oceanic rift began.

(2) As a result of the complex kinematics of the collage of tectonic plates in the Hercynian era of tectogenesis (Devonian), part of the margins of the Astrakhan–Aktobe block was pulled under northern Ustyurt, and the Tugarakchan foredeep was formed as a sedimentary barrier, located between the western and eastern segments of the basin. Transgression and simultaneous subsidence of the basin basement in the Middle–Late Devonian formed isolated carbonate seamounts (carbonate massifs) that aggraded upward. The foredeep separated both the carbonate and terrigenous basins from each other.

(3) The oblique sinistral collision of the Caspian basin and northern Ustyurt led to the complete closure of the oceanic rift, deformation of the flysch strata and the preservation of the tectonic regime at the end of the early Visean. At the same time, a young carbonate platform was formed, framing the North Ustyurt continental massif. At the same time, the differentiation of the carbonate massifs was occurring. In the seaward interior, the differentiation and further growth of carbonate massifs occurred.

(4) Late Carboniferous–Permian dextral strike–slip inversion contributed to the extensional phase of the basins of western Kazakhstan, and lateral movements along curved fault traces formed zones of local extension and compression that deformed the South Emba carbonate platform. This event in the second half of the Carboniferous period also led to the flooding of the carbonate massifs and the cessation in carbonate accumulation.

(5) The subsidence of the basin basement and the growth of island carbonate massifs led to the differentiation of the topography of the bottom of the North Caspian basin and its limited isolation from the oceanic waters. As a result, in both the Kungurian time and in the north during the Roadian time, this ensured the accumulation of salts

due to the active evaporation of sea water and its unidirectional influx from the open marine environment.

(6) In the second half of the Permian period, after the continental collision of Eastern Europe, Kazakhstan and Siberia, the island land became more active, which provided a large influx and accumulation of the powerful red-colored strata and fresh meteoric water, which gradually stopped the salt accumulation, which decreased from south to north and stretched from the Kungurian time period to Roadian time. With the accumulation of thick Upper Permian–Triassic terrigenous redbeds, salt tectonics also started and formed salt domes.

**Author Contributions:** Conceptualization, investigation, methodology, writing—original draft preparation, V.Z. and A.A.; review and editing, G.A. and V.Z.; project administration and supervision, K.I. All authors have read and agreed to the published version of the manuscript.

**Funding:** This research was funded by the Ministry of Education and Science of the Republic of Kazakhstan (project IRN, grant number AP14870515, contract number 213/30-22-24, dated 18 October 2022), and was actively supported by the leadership of the Kazakh–British Technical University (Almaty, Kazakhstan). The authors of this article are very grateful for the support that was provided.

**Data Availability Statement:** The data presented in this study are available on request from the corresponding author.

**Acknowledgments:** This research was supported by the leadership of the Kazakh–British Technical University (Almaty, Kazakhstan). The authors thank the reviewers for their valuable comments and are grateful to the editor for their careful editing.

**Conflicts of Interest:** The authors declare no conflicts of interest.

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
