# Peer review of "The Geotectonic Peculiarities of the North Caspian Permian Salt-Bearing Basins (Kazakhstan)"

_geosciences, doi:10.3390/geosciences14010023_

Round 1
Reviewer 1 Report
Comments and Suggestions for Authors
I have completed the review of your manuscript titled "Geotectonic Reasons for North Caspian Salt-bearing Sedimentary Basins (Kazakhstan) Evolution During Latest Palaeozoic." I found the topic of this manuscript to be interesting and worthy of publication in geosciences. The images included in the manuscript are visually appealing and of professional quality. However, the manuscript requires significant revisions and restructuring before it can be considered for publication. The current version appears more like a report rather than an academic paper. My main comments are as follows:
1. The abstract section needs to be rewritten. An abstract should be a concise standalone statement that provides essential information about the paper. It should present the objective, methods, results, and conclusions of the research project in a brief and non-repetitive manner. You should remove any mention of research significance and methods from the abstract. Please include a mention of the research significance in the abstract.
2. What is the scientific problem addressed in this study? In my opinion, this is the most important revision you need to consider and address. Throughout the entire text, you mainly focus on the study area without clearly stating the scientific questions. For example, in the introduction section, you should review the current research status regarding the scientific problem.
3. Building on Comment-2, you should discuss the aforementioned scientific questions.
4. In Section 4, where you discuss the Formation of the Carbonate Platform in the North Caspian Basin, please provide evidence for the existence of the Carbonate Platform, such as outcrop images.
5. The conclusion section should also be rewritten. Having 11 points is excessive! Typically, an academic paper's conclusion section should consist of 3-5 points.
Comments on the Quality of English Language
The overall English language proficiency should be significantly improved. The current text includes too many unscientific expressions. For instance, the title of the manuscript "Geotectonic Reasons for North Caspian Salt-bearing Sedimentary Basins (Kazakhstan) Evolution During Latest Palaeozoic" uses unprofessional language, such as "Geotectonic Reasons" and "Latest Palaeozoic."
Author Response
Dear Reviewer,
Thank you for your comments made to our manuscript "Geotectonic Reasons for North Caspian Salt-bearing Sedimentary Basins (Kazakhstan) Evolution During Latest Palaeozoic"
We carefully revised initial version of our manuscript and made changes and corrections as requested.
Please refer the attachment.
Regards,
A. Akhmetzhanov

Reviewer 2 Report
Comments and Suggestions for Authors
Only few corrections are indicated in the attached text

Author Response

(The authors gave the same response as above.)

Reviewer 3 Report
Comments and Suggestions for Authors
The paper is informative and is useful for revealing a regional geologic setting of salt deposition. But it is rough for the reasons listed and cannot be accepted without being in correct English and free of grammatical and syntax errors.
This paper needs an editor who can help with the many errors in English language syntax. The errors include verb tense, run-on sentences, incomplete sentences, use of direct articles (the), and more. I include here a list of lines where I found such errors. There are some I did not catch.
I suggest a less cumbersome title “Geotectonic Evolution of Latest Paleozoic North Caspian Salt-bearing Sedimentary Basins in Kazakhstan”.
A problem is the use of time vs. rock terms for geologic time scale names. I do not think that there is a consistent use of Upper, Middle and Lower for rocks, and Late, Middle145, and Early for time divisions. This needs a competent editor, and I did not get deep into this. Terms which are used on Figure 3 should be capitalized. Note Figure 3 does not use time-rock terms. For example, line 128, 129, 190-193, 401, 494, 496, 497,
The term “siliciclastic-carbonate rocks” is not defined, and it is not clear what the authors mean by this term. Is it silty carbonate? Interbedded siltstone or sandstone and carbonate? Mixed carbonate-siliciclastic turbidites? The term is used often in the text and figure captions but never defined. For example, line 81, 86, 154, 201, 209,492, 494, and many more.
The paper is well illustrated, though some figures have errors in the labelling of features. Here are some figures which need to be edited.
(Fig 4 legend has many errors).,.
Fig 5 legend has several errors. What is MOR? “Borders nude part of the Southern Urals”
Fig. 7 line 285, 286, 293,
Line 341 Rodinia formed 400 My before the end of the Neoproterozoic!
Figure 9. The figure indicates that the basement moved upwards and this is part of the locations of the salt domes. Is this justified by evidence. I missed it in the text.
Line 530. This eclogite lens is important. How is it demonstrated in the data?
In references books are cited in an awkward manner, example line 667, 671, 673, 691, 694, 718, 720, and several other places.
Misspelling in line 724
Line 718 the author is “Wilgus”
Line 734 incomplete references. Publisher?
Comments on the Quality of English LanguageSyntax or grammar errors, line 21, twice, 23, 25, 26, two errors. 37, 45 in figure caption, 60, 99, 105, 107, 108, 109, 110, 119, 166, 173, 174, 178, 182, 185-188, 200, 216, 220, 243, 244, 252, 267, 304 (the castle?), 2310, 331, 340, 352, 359, 366, 370, 376 two errors, 406 (what is a catcher), 414, 433, 502 (“left lateral” should be used instead of “left sided”, also see line 517, 599, 609,), 506, 507, 534, 535, 563, 565, 566-568-569 (spelling of Guadalupian) , 571 (bases?), 574, 613 (framed), 597 (dissected relief), 614, 623, 653,
Run on Sentences, line 10, 11, 240, 389,
Author Response

(The authors gave the same response as above.)

Round 2
Reviewer 1 Report
Comments and Suggestions for Authors
I am glad to see the major changes you have made to the revised manuscript, which have greatly improved its academic level. Thank you for your dedication and effort in revising the manuscript.
Comments on the Quality of English LanguageHowever, there are still many grammar errors in the article, making it difficult to read. I suggest you seek the help of a native English speaker or professional organization to revise the language.
Author Response
Dear Reviewer,
Thank you for your comments, made for manuscript “Geotectonic Reasons of the North Caspian Salt-bearing Sedimentary Basins (Kazakhstan) Evolution During Latest Palaeozoic”.
Regarding your comment on grammar errors which making difficult to read the article, we used English Language Editing Service (please see the attachment).
Thank you once more
Regards,
A. Akhmetzhanov

Reviewer 3 Report
Comments and Suggestions for Authors
Second set of comments on Zhemchuzhikov et al.
The title is better, but I would suggest “of the North Caspian” rather than “for”
The Abstract is clear, but it does not describe either the “model”, nor the “peculiarities”. As I understand the tectonics (Line 135 this is the important point about the eclogites and line 415), the authors attribute the differential subsidence to dense eclogites in the basement. This needs to be mentioned in the Abstract and more clearly in the conclusions (line 664 or close to that).
The abstract and the text need to more clearly describe both the peculiarities and the model.
Grammatical errors remain, lines 46, 47, 68, 109, 223, 302, 442 run on sentence
Line 301 need more explanation
Line 337 what is the “castle”?
Line 454, 458 Depressional is not used correctly. I think it refers to a site where subsidence has made a basin depression.
There is still confused use of Time vs. Rock terms “Early vs. Lower”, “Late vs. Upper”, in lines. 70 (post vs. supra-salt), 161, 185, 187, 224, 225, 331, 355, 358,
Comments on the Quality of English LanguageThere are still English errors that should be fixed.
Author Response
Dear Reviewer,
Thank you for your comments, made for manuscript “Geotectonic Reasons of the North Caspian Salt-bearing Sedimentary Basins (Kazakhstan) Evolution During Latest Palaeozoic”.
We carefully revised the body of the manuscript and made corrections to the text as requested. Regarding your comment on English grammar errors, we used English Language Editing Service.
Detailed responses to your comments are in attached file.
Regards,
A. Akhmetzhanov
